# Gut microbiota dysbiosis and neurological function recovery after intracerebral hemorrhage: an analysis of clinical samples

Yan Wang,[1] Hailong Bing,[1] Conghui Jiang,[1] Jie Wang,[1] Xuan Wang,[1] Zhengyuan Xia,[2] Qinjun Chu[1]

**ABSTRACT** We aimed to investigate the microbial community composition in patients with intracerebral hemorrhage (ICH) and its effect on prognosis. We designed two clinical cohort studies to explore the gut dysbiosis after ICH and their relationship with neurological function prognosis. First, fecal samples from patients with ICH at three time points: T1 (within 24 h of admission), T2 (3 days after surgery), and T3 (7 days after surgery), and healthy volunteers were subjected to 16S rRNA sequencing using Illumina high-throughput sequencing technology. When differential gut microbiota was identified, the correlation between clinical indicators and microbiotas was analyzed. Subsequently, the patients with ICH were categorized into GOOD and POOR groups based on their Glasgow Outcome Scale Extended (GOS-E) score, and the disparities in gut microbiota between the two groups were assessed. Univariate and multivariate logistic regression analyses were performed to identify independent risk factors. The composition and diversity of the gut microbiota in patients with ICH were different from those in the control group and changed dynamically with the extension of the course of cerebral hemorrhage. The abundances of *Enterococcaceae*, *Clostridiales incertae sedis XI*, and *Peptoniphilaceae* were significantly increased in patients with ICH, whereas *Bacteroidaceae*, *Ruminococcaceae*, *Lachnospiraceae*, and *Veillonellaceae* were significantly reduced. The relative abundance of *Enterococcus* gradually increased with the extension of the duration of ICH after surgery, and the abundance of *Bacteroides* gradually decreased. The abundance of *Enterococcus* before surgery was found to be negatively associated with patient neurological function prognosis. The original ICH score and *Lachnospiraceae* status were independent risk factors for predicting the prognosis of neurological function in patients with ICH ($P < 0.05$). Changes in the gut microbiota diversity in patients with ICH were related to prognosis. *Lachnospiraceae* may have a protective effect on prognosis.

**IMPORTANCE** Acute central nervous system injuries like hemorrhagic stroke are major global health issues. While surgical hematoma removal can alleviate brain damage, severe cases still have a high 1-month mortality rate of up to 40%. Gut microbiota significantly impacts health, and treatments like fecal microbiota transplantation (FMT) and probiotics can improve brain damage by correcting gut microbiota imbalances caused by ischemic stroke. However, few clinical studies have explored this relationship in hemorrhagic stroke. This study investigated the impact of cerebral hemorrhage on the composition of gut microbiota, and we found that *Lachnospiraceae* were the independent risk factors for poor prognosis in intracerebral hemorrhage (ICH). The findings offer potential insights for the application of FMT in patients with ICH, and it may improve the prognosis of patients.

**KEYWORDS** intracerebral hemorrhage, gut microbiota, 16S rRNA sequencing, neurological function recovery

Address correspondence to Qinjun Chu, jimmynetchu@163.com.

The authors declare no conflict of interest.

Stroke is the most common acute and critical disease in the clinical management of nervous system diseases. Intracerebral hemorrhage (ICH) is a common type of stroke. It is caused by the rupture of blood vessels, resulting in bleeding in the brain parenchyma and subarachnoid space. The prognosis of patients is poor, and the mortality rate is high (1). Surgical removal of the hematoma is the main treatment method for patients with ICH, which may reduce hematoma-related brain damage. However, surgery fails to improve the long-term prognosis of patients with ICH (2). Therefore, it is important to identify new therapeutic targets to improve the prognosis of patients with ICH.

Recently, along with the rapid growth of metagenomics, metatranscriptomics, and metaproteomics, studies have shown that the gut microbiota is closely related to the function of the central nervous system, which went by the name of the "gut-brain axis" (3, 4). Gastrointestinal dysfunction may occur after brain injury due to pathogenic factors such as damage to the structure and function of the gastrointestinal mucosal epithelial barrier, gastrointestinal motility disorders, and changes in the gut microbiota. After changes in the intestinal internal environment, endotoxins and other harmful substances in the intestine can penetrate the intestinal wall into the blood, and the damaged blood-brain barrier can exacerbate the activation of microglia, induce the aggregation of immune cells and damage-related pattern molecules to the lesion site, trigger oxidative stress, cause neuronal necrosis and apoptosis, inhibit axon regeneration, and aggravate secondary injury after brain trauma, forming a vicious cycle (5). The gut microbiota play a pivotal role in many central nervous system diseases, including Alzheimer's disease, Parkinson's disease, and ICH (6, 7). A previous study shows that ICH will quickly cause gut microbiota dysbiosis in patients with excessive growth of *Enterobacteriaceae*, thereby aggravating cerebral infarction (8). Animal experiments have reported that significant gut microbiota dysbiosis after ICH leads to neuroinflammation by affecting T-cell homeostasis, and fecal transplantation can improve neuroprognosis after ICH (9). However, at present, the relationship between ICH and gut microbiota is mostly preclinical research. Moreover, the impact of ICH on the composition of the gut microbiota and the effect of brain injury-specific microbiota changes on the recovery of neural function and prognosis after brain injury remain unknown. Clinical studies assessing changes in the gut microbiota after ICH are limited.

16S rRNA sequencing can be used to identify and classify microorganisms and provide information on the composition and diversity of microbial communities (10–12). This study aimed to explore the changes in the gut microbiota after ICH and its impact on the prognosis of patients' neurological function, in order to identify potential new treatment targets for the prognosis of patients with ICH.

## MATERIALS AND METHODS

### Study population

Patients with ICH who were admitted to the emergency department and non-hospitalized individuals who underwent health checkups during the same period were recruited in this study from May to September 2023. All participants were local residents of Henan Province. The study patients with ICH met the following inclusion criteria: age 18–75 years, admission 24 h after ICH, ICH confirmed by cranial computed tomography, and underwent hematoma evacuation; exclusion criteria included traumatic ICH, preoperative liver and kidney dysfunction, history of previous gastrointestinal surgery, history of gastrointestinal diseases, history of immune-related diseases or being given immunotherapy, receiving antibiotics or probiotics 1 month before onset, and death within 72 h after surgery.

### Fecal sample collection

Two cohorts were included in this study. Thirty patients with ICH (ICH group) and 30 healthy controls (control group) were included in the first clinical cohort. Fecal specimens

or rectal swabs (if no feces were present) from patients with ICH were collected at T1 (within 24 h after admission), T2 (3 days after surgical evacuation of the hematoma), and T3 (7 days after surgical evacuation of the hematoma). Fecal samples were collected from healthy controls during the same period. Clinical cohort 2 included 51 patients with ICH. Their fecal samples were collected within 24 h after admission but before surgery. All specimens were collected using sterile sample boxes and immediately frozen at −80℃.

## Data collection

The patients' demographic and clinical information was documented, including age, sex, underlying diseases, Glasgow Coma Scale (GCS) score, original intracerebral hemorrhage (OICH) score, time from onset to surgery, operation time, bleeding volume, intensive care unit (ICU) mechanical ventilation time, ICU stay time, National Institutes of Health Stroke Scale (NIHSS) score at admission, 7 and 14 days after surgery, and the Extended Glasgow Outcome Scale (GOS-E) score 30 days after surgery. The OICH score (Table 1) is the most extensively validated clinical grading scale in cerebral hemorrhage patients. The score ranges from 0 to 6 and is a valid clinical prediction rule for short-term mortality in ICH patients (13, 14). The GOS-E score (Table 2) stratifies patients as follows: (i) death, (ii) vegetative state, (iii) lower severe disability, (iv) upper severe disability, (v) lower moderate disability, (vi) upper moderate disability, (vii) lower good recovery, and (viii) upper good recovery (15). Higher GOS-E scores indicate a more favorable neurological outcome than do lower scores; an unfavorable neurological outcome was defined as a GOS-E score less than 5 (16).The ICH patients in the second clinical cohort were divided into the POOR group (score < 5) or GOOD group (score ≥ 5), according to the GOS-E score of patients with ICH 30 days after surgery.

In this study, the primary outcome was to analyze the difference of gut microbiota between healthy control subjects and patients with ICH at T1, T2, and T3 time points. And, the correlation between the difference of gut microbiota and the prognosis of neurological function was calculated. The secondary outcome was to compare the differences in gut microbiota within 24 h after ICH between the POOR and GOOD groups, and the correlation between the differences in gut microbiota and the prognosis of neurological function was calculated.

## Genomic DNA extraction, amplification, and sequencing

Total community genomic DNA was extracted using an E.Z.N.A. MagBind Soil DNA Kit (Omega, M5635-02). Purity was determined using a Qubit 4.0 (Thermo, USA), and high-quality DNA was amplified using the 341F (5′-CCTACGGGNGGCWGCAG-3′) and 805R (5′-GACTACHVGGGTATCTAATCC-3′) primers in the hypervariable V3–V4 region of the bacterial 16S rRNA gene. PCR mixtures were prepared with 1 µL of each primer and 2 µL of template DNA; 2× Hieff Robust PCR Master Mix (Yeasen, 10105ES03, China) was added to a final volume of 30 µL. The plate was sealed, and PCR was performed using a thermal instrument (Applied Biosystems 9700, USA). High-quality PCR products were sequenced on an Illumina MiSeq platform by Sangon BioTech (Shanghai, China).

## Processing of sequencing data

We used the PEAR software (version 0.9.8) to assemble two short Illumina readings depending on the overlap. FASTA and QUAL files were individually generated from the FASTQ files. Usearch software (version 11.0.667) was used to cluster the valid tags into operational taxonomic units (OTUs) with similarity ≥ 97%. Chimeric sequences and singleton OTUs were removed, after which the remaining sequences were sorted into each sample based on the OTUs. The tag sequence with the highest abundance was selected as a representative sequence within each cluster. Bacterial and fungal OTU representative sequences were classified taxonomically by blasting against the Ribosomal Database Project database and UNITE fungal ITS database, respectively.

**TABLE 1** The original ICH score

| Component | OICH score points |
|---|---|
| GCS score | |
| 3–4 | 2 |
| 5–12 | 1 |
| 13–15 | 0 |
| ICH volume, $cm^3$ | |
| ≥30 | 1 |
| <30 | 0 |
| Intraventricular hemorrhage | |
| Yes | 1 |
| No | 0 |
| Infratentorial origin of ICH | |
| Yes | 1 |
| No | 0 |
| Age, yr | |
| ≥80 | 1 |
| <80 | 0 |
| Total OICH score | 0–6 |

## Statistical analysis

Data were analyzed using the SPSS 25 statistical software. Categorical variables are presented as numbers and percentages, comparisons between groups were performed with $\chi^2$ test, and the normal distribution data were compared with the t test and expressed as mean and standard deviation (SD). The Wilcoxon rank-sum test was used to compare the non-normally distributed measured data between groups; the results were expressed as medians and interquartile range (IQR). The sparse curve of the observed number of OTUs was constructed, and all α-diversity indexes were calculated using Mothur software (version 3.8.31). ANOVA test was used to compare the multiple groups. Bacterial diversity of each sample, principal coordinates analysis (PCoA) based on Bray-Curtis differences, was used to evaluate β-diversity between samples, and permutation multivariate analysis of variance (PERMANOVA) was used to test for significant differences between groups. Univariate and multivariate logistic regression analyses were performed to determine the risk factors for the prognosis of cerebral

**TABLE 2** Overview of the categories of the GOS-E score[a]

| GOSE eight-point scale | Domain | Criteria |
|---|---|---|
| Dead | | |
| Vegetative state | Consciousness | |
| Lower severe disability (conscious but dependent) | Function in home | Unable to look after themselves for 8 h |
| Upper severe disability | Function in home, function outside the home | Unable to look after themselves for 24 h OR unable to shop OR unable to travel |
| Lower moderate disability (independent but with limitations in one or more activities) | Work/study, social and leisure activities, family and friendships | Unable to work/study OR unable to participate OR constant problems |
| Upper moderate disability | Work, social and leisure activities, family and friendships | Reduced work capacity OR participate much less OR frequent problems |
| Lower good recovery (return to normal life) | Social and leisure activities, family and friendships, symptoms | Participate a bit less OR occasional problems OR some symptoms affecting daily life |
| Upper good recovery | | No problems |

[a]GOS-E, Glasgow Outcome Scale-Extended.

hemorrhage and neurological dysfunction, and odds ratios and 95% confidence intervals were calculated. We used the drawing function of R software to draw the forest map. The pheatmap software package (version 1.0.12) was used to draw the heat map.

## RESULTS

### Clinical characteristics in the first cohort

A total of 30 patients with ICH and 30 controls were included in this study; their clinical characteristics are shown in Table 3. Compared with the control group, the proportion of patients with ICH with hypertension was higher (73% vs. 30%, $P = 0.002$).

### Gut microbiome characterization in ICH and control groups

The relative abundances of the most abundant bacterial phyla and families of microbiota in the two groups are shown in Fig. 1A and B. The most abundant phyla were *Bacillota*, *Bacteroidota*, *Pseudomonadota*, and *Actinomycetota* (Fig. 1A). In addition, at the level of family, the abundances of *Enterococcaceae*, *Clostridiales incertae sedis XI*, and *Peptoniphilaceae* were significantly increased in patients with ICH, whereas *Bacteroidaceae*, *Ruminococcaceae*, *Lachnospiraceae*, and *Veillonellaceae* were significantly reduced (Fig. 1B; Fig. S1).

α-Diversity showed that comparison of patients with ICH at T3 and T1, T2, and control groups and differences of α-diversity have statistical significance (Fig. 2A and B).

PCoA was used to determine whether there were significant differences in the microbiota structure between ICH patients and controls. The microbial composition of the ICH group in different time points (T1, T2, and T3) was significantly different from that of control group according to Bray-Curtis distance, PERMANOVA test (Fig. 2C) (T1 vs. T2, $R^2 = 0.03$, $P = 0.06$; T1 vs. T3, $R^2 = 0.10$, $P < 0.001$; T1 vs. Con, $R^2 = 0.11$, $P < 0.001$; T2 vs. T3, $R^2 = 0.04$, $P < 0.05$; T2 vs. Con, $R^2 = 0.09$, $P < 0.001$; and T3 vs. Con, $R^2 = 0.15$, $P < 0.001$). The richness and diversity of gut microbiota in patients with ICH were different from

**TABLE 3** Clinical characteristics of the ICH and control groups[a]

| Characteristic | ICH group ($n = 30$) | Control group ($n = 30$) | $P$-value |
|---|---|---|---|
| Age (yr), mean (SD) | 57.7 (13.4) | 56.1 (10.5) | 0.60 |
| Height (cm), mean (SD) | 166.1 (8.8) | 162.4 (8.2) | 0.10 |
| Weight (kg), mean (SD) | 70.0 (13.4) | 65.9 (11.9) | 0.74 |
| Gender, $n$/total $N$ (%) | | | 0.30 |
| Male | 16/30 (53%) | 11/30 (37%) | |
| Female | 14/30 (47%) | 19/30 (63%) | |
| Smoker | 8/30 (27%) | 6/30 (20%) | 0.76 |
| Alcoholism | 2/30 (7%) | 2/30 (7%) | 1 |
| Hypertension, $n$/total $N$ (%) | 22/30 (73%) | 9/30 (30%) | 0.002[b] |
| DM, $n$/total $N$ (%) | 5/30 (17%) | 5/30 (17%) | 1 |
| CHD, $n$/total $N$ (%) | 6/30 (20%) | 5/30 (17%) | 0.74 |
| GCS score on admission, median (IQR) | 5 (4, 8) | | |
| OICH score on admission, median (IQR) | 2 (1, 3) | | |
| NIHSS score on admission, median (IQR) | 29 (29, 29) | | |
| NIHSS score at 7 days, median (IQR) | 27 (7, 29) | | |
| NIHSS score at 14 days, median (IQR) | 17 (3, 29) | | |
| GOSE median (IQR) | 4 (2,6) | | |
| ICU-LOS (days), median (IQR) | 8 (5,18) | | |

[a]SD, standard deviation; IQR, interquartile range; DM, diabetes mellitus; CHD, coronary heart disease; GCS, Glasgow Coma Scale; OICH: original intracerebral hemorrhage; NIHSS, National Institutes of Health Stroke Scale; ICU, intensive care unit; LOS, length of stay.
[b]$P < 0.05$.

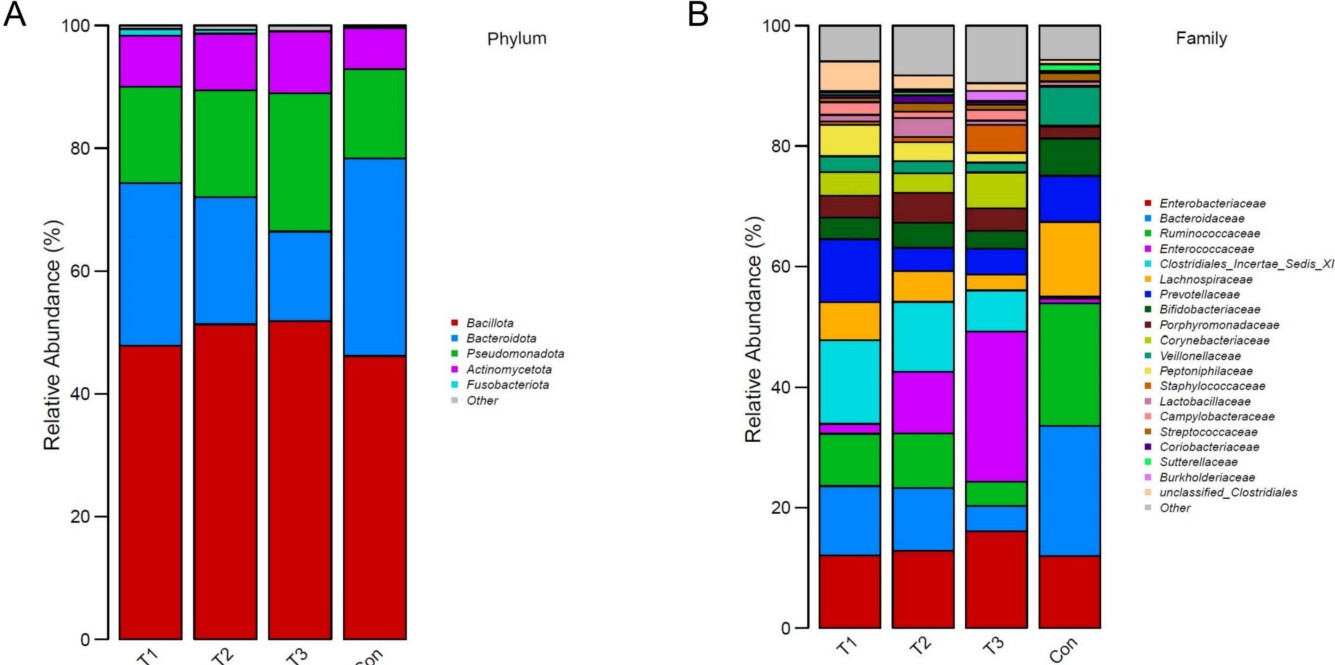

**FIG 1** The gut microbial composition. The relative abundance of bacteria at the phylum level (A) and family level (B).

those in the control group and changed dynamically with the extension of the course of the disease and prolongation of the course of ICH.

Linear discriminant analysis effect size (LEfSe) was used to reveal significant bacterial differences in gut microbiota among the different phases of the ICH and control groups at the genus level, and the results revealed that *Bacteroides*, *Faecalibacterium*, *Ruminococcus*, *Roseburia*, *Gemmiger*, and *Lachnospiraceae* were significantly overrepresented in the control group. *Prevotella*, *Peptoniphilus*, *Porphyromonas*, and *Campylobacter* were all significantly overrepresented in patients with ICH at T1, whereas *Enterococcus*, *Corynebacterium*, *Staphylococcus*, *Ralstonia*, and *Phenylobacterium* were all significantly overrepresented in patients with ICH at T3 (Fig. 3A and B). In addition, the results showed that, compared with that in the control group, the relative abundance of *Enterococcus* gradually increased with the extension of the duration of ICH after surgery, reaching its highest at T3, and the abundance of *Bacteroides* gradually decreased, reaching its lowest at T3 (Fig. 3C through E).

## Correlation analysis between significantly different microbial communities and clinical indicators

We analyzed the correlation between 15 bacterial groups with significant differences in LEfSe analysis and clinical indicators and found that *Enterococcus* was positively correlated with the NIHSS score on days 7 and 14 after surgery and was negatively correlated with the GOS-E prognostic score, while *Prevotella* was positively correlated with preoperative and postoperative NIHSS scores and ICU stay time at T1 (Fig. 4). These results indicated that changes in *Enterococcus* and *Prevotella* may be closely correlated with patient prognosis.

## Clinical characteristics in the second cohort

A total of 51 patients with ICH were included in this study; the patients were divided into the POOR group (score < 5, *n* = 28) or GOOD group (score ≥ 5, *n* = 23), according to the GOS-E score of patients with ICH 30 days after surgery. The clinical characteristics of the patients are shown in Table 4. Compared with the POOR group, high-density lipoprotein

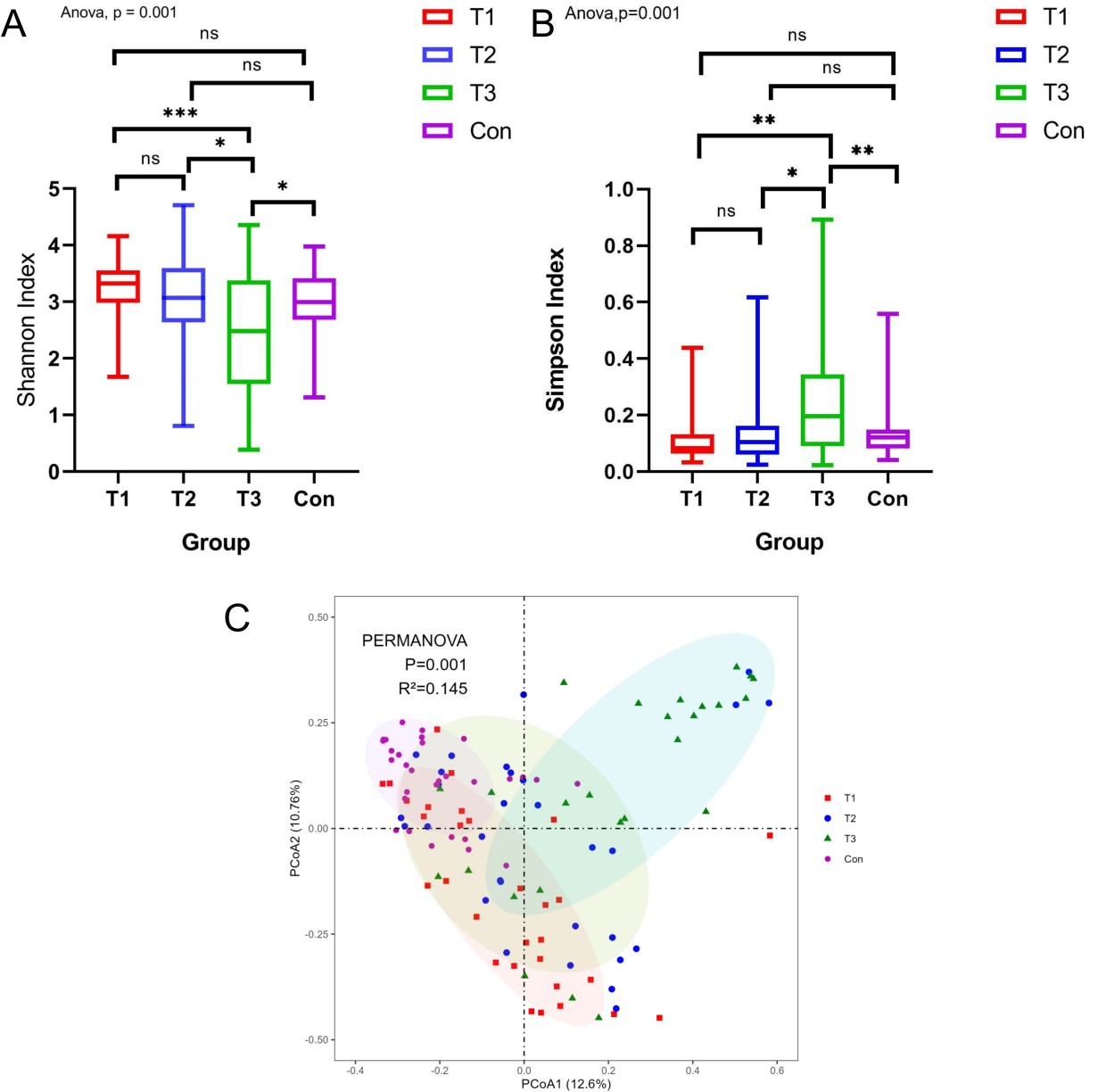

**FIG 2**   Change in bacterial diversity between the ICH and control groups. (A) Simpson index. (B) Shannon index. (C) PCoA showing the gut microbiota composition among healthy controls and T1, T2, and T3 phases of patients with ICH. *$P < 0.05$, **$P < 0.01$, and ***$P < 0.001$.

cholesterol and GCS on admission were significantly higher in the GOOD group ($P < 0.05$), while OICH on admission, NIHSS score at 7 days, and length of stay in the ICU were significantly lower in the GOOD group ($P < 0.05$).

## Gut microbiome characterization in patients with ICH between the POOR and GOOD groups

We compared α-diversity analyses (Shannon index) between the POOR group and the GOOD group (Fig. 5A). Next, we performed β-diversity analysis PCoA analysis was performed based on Bray-Curtis distance, and PERMANOVA test was performed to compare the results of the PCoA differences between groups ($R^2 = 0.02$, $P = 0.67$; Fig. 5B). The results showed that there were no significant difference in the α-diversity and β-diversity between the two groups, suggesting that species richness, evenness, and diversity of the gut microbiome were similar between the two groups.

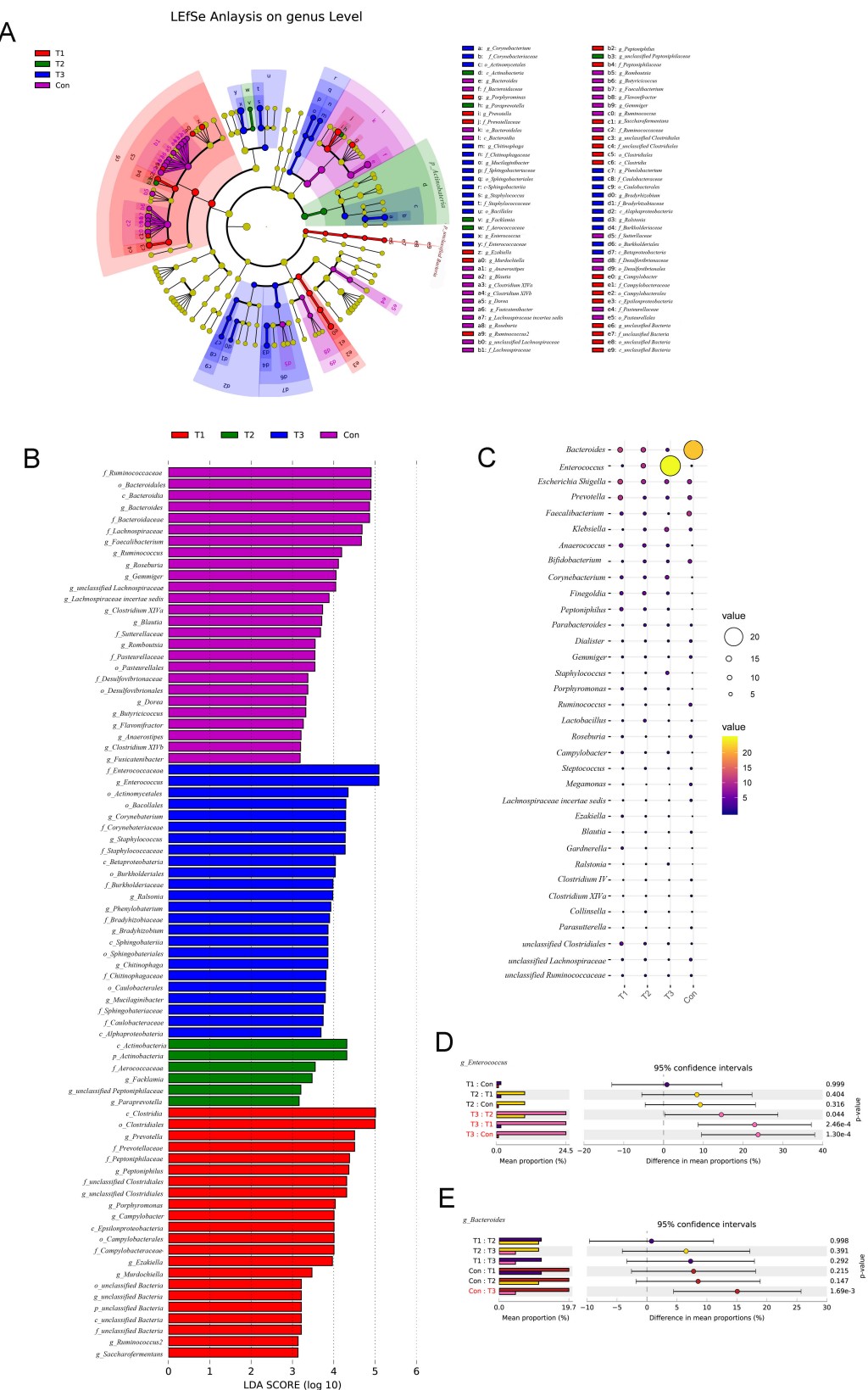

**FIG 3** Taxonomic differences of microbiota in the ICH and control groups. (A and B) LEfSe revealed significant bacterial differences in microbiota among different phases of the ICH and control groups. Linear discriminant analysis (LDA) scores (log10) > 2 (B). (C) Ballon of gut microbial composition at the genus level. (D) The abundance of *Enterococcus* between the ICH and control groups. (E) The abundance of *Bacteroides* between the ICH and control groups.

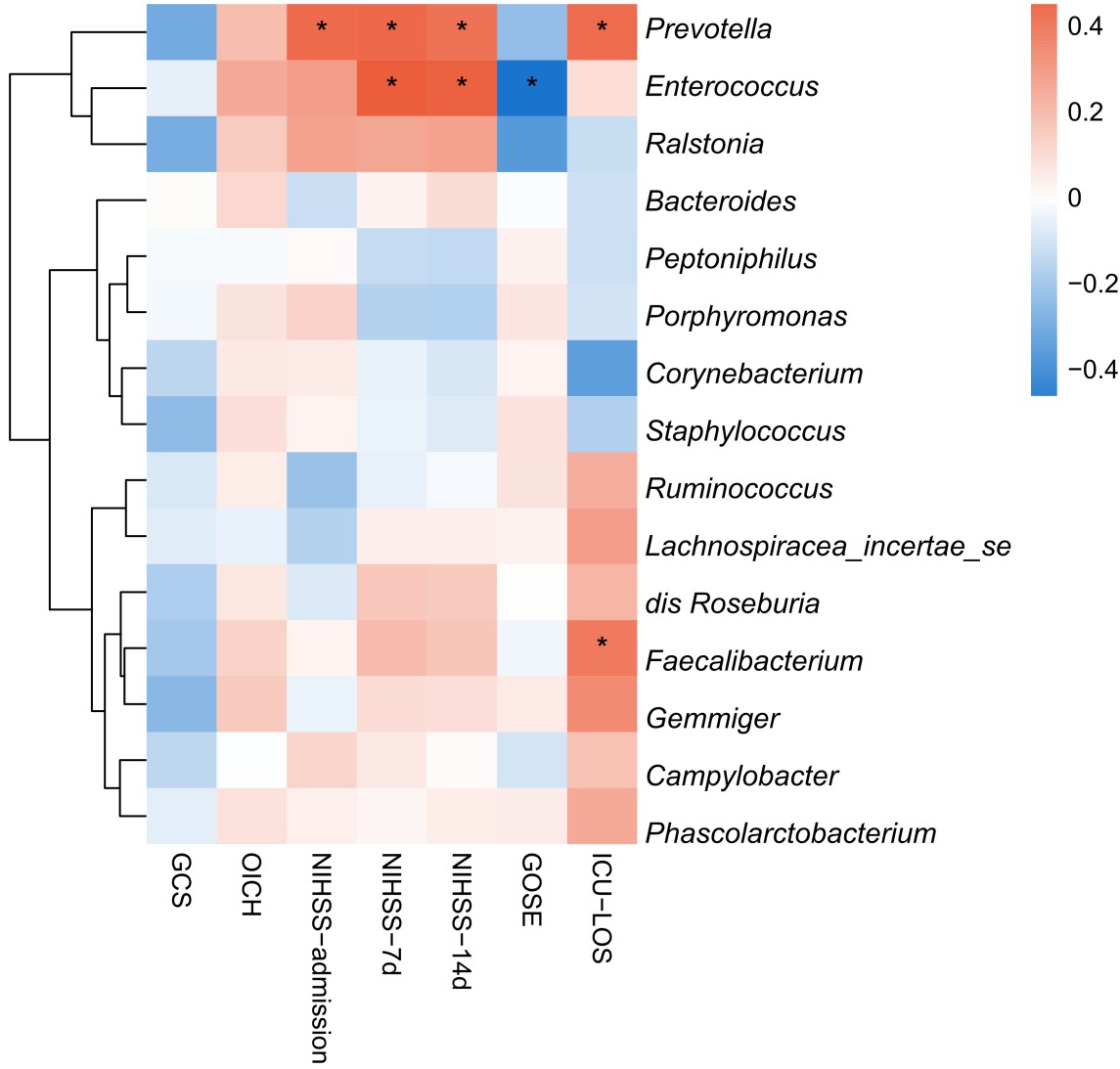

**FIG 4** Heatmap of Spearman's rank correlation coefficient among clinical indicators and 15 genera with LDA scores (log10) > 4. *$P < 0.05$.

LEfSe was used to reveal significant bacterial differences in gut microbiota between the POOR and GOOD groups at the genus level, and the results revealed that *Enterococcus* was significantly overrepresented in the POOR group, whereas *Coprobacillus*, *Lachnospiraceae*, and *Pyramidobacter* were significantly overrepresented in the GOOD group (Fig. 5C and D).

### Independent risk factors for poor prognosis in patients with ICH

Univariate and multivariate logistic regression analyses were performed to identify independent risk factors for poor prognosis in ICH. OICH and *Lachnospiraceae* were independent risk factors for poor prognosis in ICH ($P < 0.05$; Fig. 6A and B).

### DISCUSSION

This study found that the richness and diversity of gut microbiota in patients with ICH were different from that in healthy individuals. The intestinal microorganisms in patients with ICH were significantly enriched at the family level and consisted of *Enterococcaceae*, *Clostridiales incertae sedis XI*, and *Peptoniphilaceae*. *Bacteroidaceae*, *Ruminococcaceae*, *Lachnospiraceae*, and *Veillonellaceae* significantly decreased. Upon conducting an analysis of the gut microbiota composition, we discerned significant modifications in the

**TABLE 4** Clinical characteristics of the POOR and GOOD groups[a]

| Characteristic | GOOD group (N = 23) | POOR group (N = 28) | P-value |
|---|---|---|---|
| Age (yr), mean (SD) | 54.8 (16.0) | 53.4 (17.3) | 0.77 |
| BMI (kg/m$^2$), mean (SD) | 25.0 (4.3) | 24.1 (3.6) | 0.42 |
| Gender, n/total N (%) | | | 0.24 |
| Male | 16/23 (69.6%) | 15/28 (53.6%) | |
| Female | 7/23 (30.4%) | 13/28 (46.4%) | |
| Smoker, n/total N (%) | 3/23 (13.0%) | 7/28 (25.0%) | 0.47 |
| Alcoholism, n/total N (%) | 0/23 (0%) | 3/28 (10.7%) | 0.31 |
| Hypertension, n/total N (%) | 11/23 (47.8%) | 18/28 (64.3%) | 0.37 |
| DM, n/total N (%) | 4/23 (17.4%) | 5/28 (17.9%) | 0.97 |
| CHD, n/total N (%) | 4/23 (17.4%) | 4/28 (14.3%) | 0.76 |
| Preoperative SBP (mmHg), mean (SD) | 156 (26.2) | 167 (41.1) | 0.27 |
| Preoperative DBP, mean (SD) | 87.3 (14.4) | 92 (23.1) | 0.38 |
| WBC, ×109/L, median (IQR) | 10.4 (8.9, 11.4) | 11.4 (8.9, 13.3) | 0.11 |
| Triglyceride, mmol/L, median (IQR) | 1.0 (0.6, 2.0) | 1.5 (0.8, 2.6) | 0.59 |
| Glucose, mmol/L, median (IQR) | 8.5 (7.0, 10.8) | 9.5 (7.5, 12.5) | 0.29 |
| HDL-C, mmol/L, mean (SD) | 1.0 (0.3) | 0.8 (0.3) | 0.04[b] |
| LDL-C, mmol/L, mean (SD) | 1.8 (0.7) | 1.8 (0.8) | 0.78 |
| Creatinine on admission, μmol/L, median (IQR) | 58.8 (45.4, 68.3) | 52.6 (42.3, 83.3) | 0.99 |
| Uric acid, μmol/L, median (IQR) | 297.0 (210.6, 379.2) | 262.0 (164.0, 435.9) | 0.89 |
| Minimum intraoperative SBP (mmHg), mean (SD) | 106.4 (9.2) | 99 (17.8) | 0.10 |
| Minimum intraoperative DBP (mmHg), mean (SD) | 61.3 (8.0) | 54.8 (13.9) | 0.05 |
| Creatinine at 1 day, μmol/L, median (IQR) | 52.5 (42.1, 74.5) | 54.7 (39.7, 77.3) | 0.93 |
| GCS score on admission, median (IQR) | 8 (5, 14) | 5 (3, 6) | 0.001[b] |
| OICH score on admission, median (IQR) | 1 (1, 2) | 2 (2, 3) | 0.001[b] |
| NIHSS score on admission, median (IQR) | 29 (10, 29) | 29 (29, 29) | 0.006[b] |
| NIHSS score at 7 days, median (IQR) | 5 (3, 9) | 29 (27.2, 29) | <0.001[b] |
| Operation time (min), median (IQR) | 155 (90, 215) | 160 (85, 237.5) | 0.59 |
| Bleeding (mL), median (IQR) | 200 (20, 400) | 300 (27.5, 475) | 0.40 |
| ICU-LOS (days), median (IQR) | 4 (2, 8) | 11.5 (8, 15.8) | <0.001[b] |

[a]SD, standard deviation; IQR, interquartile range; BMI, body mass index; DM, diabetes mellitus; CHD, coronary heart disease; SBP, systolic blood pressure on admission; DBP, diastolic blood pressure on admission; WBC, white blood cell; TG, triglyceride; LDL-C, low-density lipoprotein cholesterol; HDL-C, high-density lipoprotein cholesterol; GCS, Glasgow Coma Scale; OICH: original intracerebral hemorrhage; NIHSS, National Institutes of Health Stroke Scale; ICU, intensive care unit; LOS, length of stay.
[b]$P < 0.05$.

bacterial genera of patients with ICH compared with the control group, using the LEfSe algorithm. It is noteworthy that as the ICH patients advanced after surgery, there was a gradual increase in the abundances of *Enterococcus*. Conversely, *Bacteroides* were found to be enriched in the healthy control population. The abundance of *Enterococcus* before surgery was negatively correlated with the GOS-E score at 30 days after surgery. The prognosis of neurological function in patients with ICH was related to the preoperative OICH score and *Lachnospiraceae* abundance. Compared with patients with ICH with poor neurological function prognosis, patients with better postoperative recovery had a higher abundance of *Lachnospiraceae* at admission, indicating that intestinal *Lachnospiraceae* may have a protective effect on patients with ICH. This study establishes a foundation for identifying potential prevention and treatment strategies involving gut microbiota transplantation for individuals afflicted with ICH. Future studies will be required to explore the mechanism of *Lachnospiraceae* in enhancing the prognosis of patients with ICH.

Spontaneous cerebral hemorrhage is a serious nervous system disease characterized by rapid onset, rapid progression, and high mortality and disability rates. The mortality

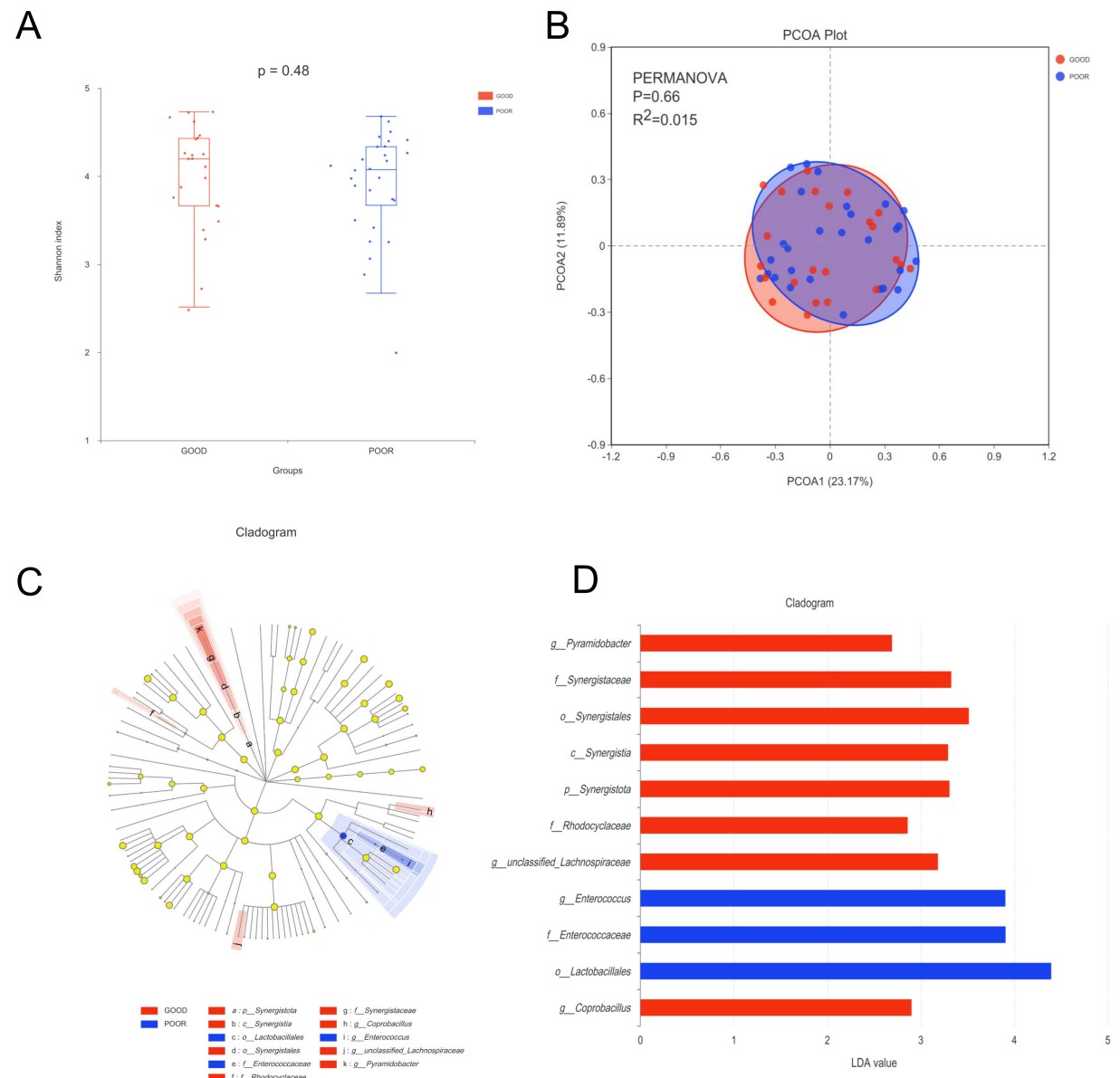

**FIG 5** Change in bacterial diversity and taxonomic differences between the POOR and GOOD groups. (A) Shannon index. (B) PERMANOVA test. (C and D) LEfSe revealed significant bacterial differences in microbiota among different phases of the POOR and GOOD groups.

rate of cerebral hemorrhage is approximately 40% in 1 month. Approximately 12% to 39% of the patients achieved long-term functional independence (17). In recent years, great progress has been made in understanding the etiology, pathophysiology, acute treatment, and prevention of cerebral hemorrhage. Rapid acute intervention measures (drug therapy and minimally invasive surgery) may improve acute prognosis (18). Adverse events with long-term prognosis would have a significant impact on the rehabilitation of patients, including cognitive disorders, mental disorders, seizures, recurrent cerebral hemorrhage, and subsequent thromboembolic events (19). These events reduce the quality of life of patients and impose a huge burden on them, their caregivers, their family members, and society (20, 21).

The gut microbiota, which is closely related to human health, is the largest microbial community in the human body. The changes in gut microbiota can produce various metabolites (neurotransmitters, neuropeptides, reactive nitrogen substances, and others) (22), affecting physiological, behavioral, and cognitive functions through the

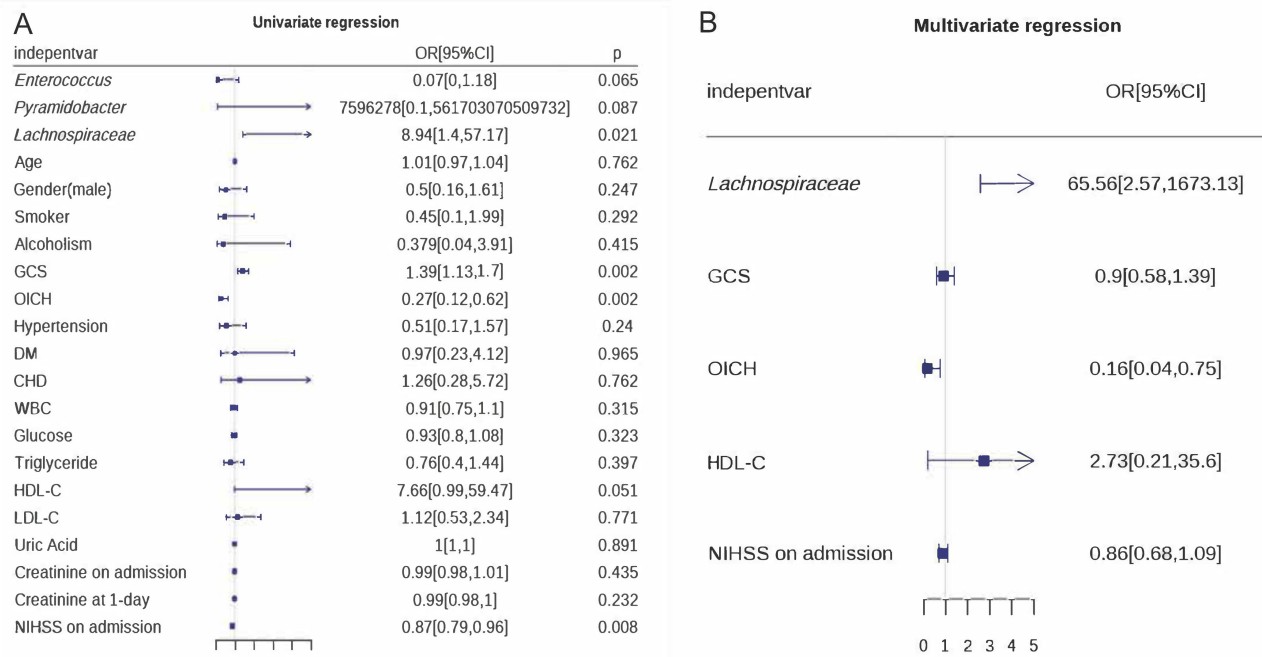

**FIG 6** Independent risk factors for poor prognosis in ICH. Univariate (A) and multivariate logistic regression (B) analyses performed to identify the independent risk factors for poor prognosis in ICH.

gut-brain axis (23). A growing body of evidence supports the role of the gut microbiota in stroke, stroke prognosis, and recovery (24). The gut microbiota can increase the risk of a cerebrovascular event, playing a role in the onset of stroke. Conversely, stroke can induce dysbiosis of the gut microbiota and epithelial barrier integrity (8) . But few studies have been performed to reveal the changes of the gut microbiota associated with ICH, so we aimed to explore the changes in the gut microbiota in ICH patients and their impact on the prognosis of patients' neurological function.

In this study, the proportion of patients with a history of hypertension was significantly higher in the ICH group than in the control group. Li et al. showed that the types and quantities of gut microbiota in patients with hypertension and prehypertension were lower than those in the normal population. However, *Klebsiella* and *Pruisnia* were significantly increased, indicating that the gut microbiota of patients may be closely related to the development of hypertension (25). Patients with gut microbiota disorders are prone to obstructive sleep apnea, which leads to the occurrence of hypertension, and long-term hypertension will increase the chance of ICH (26). In addition, our study revealed changes in the richness and diversity of the gut microbiota in patients with ICH. The results of these studies are similar to those of a previous study that showed that the gut microbiota may change during cerebral hemorrhage (27). Cerebral hemorrhage can induce apoptosis, oxidative stress, and central and peripheral immune inflammatory reactions and change the abundance and composition of the gut microbiota. However, reduced diversity and microbial overgrowth further lead to decreased intestinal motility and increased permeability. Pro-inflammatory T cells can migrate into the brain, increase vascular permeability, and secrete pro-inflammatory factors that aggravate inflammatory brain injury (9).

Patients with ICH had specific gut microbiota characteristics, including significant changes in *Mycobacterium verrucosum*, a decrease in *Cladosporium*, and an increase in *Bacteroides* (28). In our study, compared with the control group, the relative abundance of *Enterococcus* gradually increased with the extension of the duration of ICH after surgery, reaching the highest level at T3, and the abundance of *Bacteroides* gradually decreased, reaching its lowest level at T3. The abundance of *Bacteroides* in

the gut microbiota of patients with neurological diseases is controversial. Zang et al. explored the gut microbiota in various neurological and psychiatric disorders, and their results showed that at the genus level, *Bacteroides* was increased in attention deficit hyperactivity disorder, anorexia nervosa, generalized anxiety disorder , and spinal cord injury patients (29). But the acute ischemic stroke patients with higher stroke severity seemed to have lower abundances of short-chain fatty acid (SCFA)-producing bacteria (*Bacteroides*, *Lachnospiraceae*, *Faecalibacterium*, *Blautia*, and *Anaerostipes*) and higher abundances of *Lactobacillaceae*, *Akkermansia*, *Enterobacteriaceae*, and *Porphyromonada-ceae* (30). The other prospective case-control study found that the gut microbiota was more vulnerable to disruption and decreased SCFA-producing bacteria in patients with ischemic and hemorrhagic stroke compared with non-stroke-matched controls (31). It is difficult to say whether *Bacteroides* have a negative or positive effect on the host. *Bacteroides* are able to benefit the host by preventing infection with potential pathogens that may colonize and infect the gut. However, *Bacteroides*-derived metabolites, carboxylic acids and monosaccharides, may also cause damage to the host (32). Further studies are needed to explore the role of *Bacteroides* in the gut microbiota and brain-gut axis in ICH patients.

*Enterococcus*, belonging to the *Bacillota* phylum, is regarded as a commensal organism in the human gastrointestinal tract (33). The enriched intestinal *Enterobacteria-ceae* may be related to the elevated inflammatory response or infections poststroke, which are associated with severe brain injury and poor stroke outcome. Clinical trials of patients with stroke (30, 34, 35) showed that the proportions of *Bacillota* and *Bacter-oidota* change owing to intestinal ecological imbalance. The abundance of opportun-istic pathogens (*Megasphaera*, *Enterobacter*, and *Desulfovibrio*) increased, whereas the abundance of beneficial short-chain fatty acid-producing bacteria (*Blautia*, *Roseburia*, *Anaerostipes*, *Bacteroides*, *Lachnospiraceae*, and *Faecalibacterium*) decreased. In patients with traumatic cerebral hemorrhage, the relative abundance of *Enterococcus*, *Bacteroides*, *Akkermansia*, and *Lachnoclostridium* also increased, indicating that changes in *Enterococ-cus* might play a role in the development of brain injury (36). Additionally, *Enterococcus* are an important clinical cause of bloodstream infection (37). Luo et al. found that gut dysbiosis with enriched *Enterococcus* increased the risk of ICH and subsequently stroke-associated pneumonia (27). In our study, Enterococcus was negatively correlated with the GOS-E prognostic score at T1, and the abundance of *Enterococcus* in the gut microbiota of patients with poor prognosis of middle nervous system function in the second cohort of this study was also higher than that in the GOOD group. Therefore, *Enterococcus* may be related to the aggravation of the disease in patients with ICH. Our research also discovered a positive correlation between *Prevotella* and both preopera-tive and postoperative NIHSS scores, as well as the duration of ICU stay. However, no significant correlation was found between *Prevotella* and the GOS-E score. *Prevotella* is typically associated with a diet rich in plants, carbohydrates, and fiber. The inflam-mation of the mucosa mediated by *Prevotella* can lead to the systemic dissemination of inflammatory mediators, bacteria, and bacterial products, which in turn may affect systemic disease outcomes (38).

Identifying the risk factors for poor prognosis in patients with cerebral hemorrhage is important for clinical management. There are many risk factors for a poor progno-sis in patients with ICH, including age, sex, hypertension, smoking, excessive drink-ing, low-density lipoprotein cholesterol, low triglycerides, and oral anticoagulants (17). However, these risk factors are inconsistent with those reported previously. In this study, patients with a worse neurologic outcome (lower GOSE scores at 30 days after surgery) had lower high-density lipoprotein, lower GCS scores, and higher OICH and NIHSS scores at admission. It is well known that clinical scoring tools such as GCS, OICH, and NIHSS are used to evaluate neurological function and prognosis in patients with ICH. Lower HDL may be both an indicator of poor health status that must be accounted for in measure-ments of frailty and/or has a direct pathophysiological effect on cerebral vasculature (39). We included the factors associated with the prognosis of ICH patients and the

differential gut microbiota between the POOR and GOOD groups in the univariate regression analysis. To avoid the effects of intraoperative hypotension on renal function, gut microbiota changes, and neurological outcome, we included the creatinine levels of patients before and after surgery in the two groups of patients. Among all the collected clinical data, the GCS score and NIHSS score were significant in the univariate logistic regression analysis but not significant in the multivariate logistic regression analysis. Conversely, the OICH score and *Lachnospiraceae* enrichment remained significant in both the univariate and multivariate logistic regression analyses, indicating that the OICH score and *Lachnospiraceae* were independent risk factors for poor prognosis in ICH. The higher the OICH score (OR 0.16, 95% CI 0.04 to 0.75, $P = 0.01$), the lower the abundance of *Lachnospiraceae* (OR 65.56, 95% CI 2.57 to 1,673.13, $P = 0.02$) and the worse the outcomes of neurological function in patients with ICH.

Clinical grading scales play an important role in the evaluation and management of patients with acute neurological disorders, and the OICH score is a simple clinical grading scale that allows risk stratification on presentation with ICH. Gregório et al. have revealed that the OICH score is a valid clinical predictive tool for short-term mortality in patients with ICH and 30-day mortality increased steadily with the original ICH Score (13, 14). Appelboom et al. have illustrated that the OICH is a valid clinical grading scale with high predictive accuracy for functional outcomes after arteriovenous malformation-associated ICH (40). Usefulness of the OICH score in predicting the 30-day outcome is confirmed in our cohort of ICH patients. The *Lachnospiraceae* family holds promise as a source of next-generation probiotics. Many species within *Lachnospiraceae* contribute important functions, such as SCFA production and antibiotic production in the human gastrointestinal tract (41). A study on ischemic stroke showed that *Lachnospiraceae* (genus) had a significant protective effect against ischemic stroke (42, 43). In addition, Shen et al. found that *Lachnospiraceae UCG008* (genus) is a potential risk factor for hemorrhagic stroke (44). *Lachnospiraceae UCG008* (genus) may also contribute to ICH through an inflammatory response (T and NK cells) (45). Thus, the results of these studies are consistent with those of this study. The *Lachnospiraceae* family contains abundant obligate anaerobic microorganisms that produce short-chain fatty acids and secondary bile acids in the human intestine, which are important for maintaining intestinal integrity, microbial homeostasis, immune function, and energy balance (46). Sodium butyrate is a short-chain fatty acid and a strong histone deacetylation inhibitor. Many studies have shown that it is beneficial to increase histone acetylation to improve the transcriptional activity of genes (47). In animal models of stroke (48), Parkinson's disease (49), and Huntington's disease (50), butyrate has been found to promote the acetylation of histones, which could reduce pathological damage. This indicates that the neuroprotective effect of butyrate is partly achieved by promoting histone acetylation.

However, this study had some limitations. First, we collected stool samples from healthy individuals as controls instead of from premorbid individuals to determine the unpredictability of ICH. There may be mixed factors, such as basic diseases, diet, and living habits. In this study, the participants were from the same area, and the acquisition time was the same season, which reduced the impact of time and space. Second, antibiotics are inevitable and important factors in patients with ICH and gut microbiota. The effects of different types of antibiotics and their use on the gut microbiota *in vivo* are not clear. We selected stool samples from patients with ICH within 24 h after admission to analyze the correlation of prognosis with a view to reducing the influence of antibiotics on the gut microbiota. Finally, this study found a correlation between gut microbiota and cerebral hemorrhage. A large number of samples and further metabolomics analysis could be used to explain the causal relationship and mechanism of gut microbiota and cerebral hemorrhage and to verify the relationship between them. The effect of implanted flora on the prognosis of spontaneous cerebral hemorrhage has been demonstrated in animal experiments.

## Conclusion

This study found significant changes in the abundance and diversity of gut microbiota in ICH patients compared with healthy individuals. *Enterococcus* was related to the prognosis of neurological function; the original ICH score and *Lachnospiraceae* status were independent risk factors for predicting the prognosis of neurological function in patients with ICH. Relatively few clinical studies have been conducted on the relationship between neurological function and gut microbiota, and we did not directly explain the causal relationship between them. However, this study provides basic information on the bacterial diversity and composition of this rich population. The gut microbiota is a potential therapeutic target for patients with ICH, and fecal microbiota transplantation may improve individual outcomes and quality of life.

## ACKNOWLEDGMENTS

We would like to express our gratitude to our co-workers and all the participants in this study.

Q.J.C. and Y.W. conceived and designed the study; Y.W and H.L.B. wrote the manuscript; H.L.B., J.W., and C.H.J. performed the experiments. Y.W. and X.W. analyzed the data. Q.J.C. and Z.Y.X. revised the manuscript. All authors have read and agreed to the finalized version of the manuscript.

## AUTHOR AFFILIATIONS

[1]Department of Anesthesiology and Perioperative Medicine, Zhengzhou central Hospital Affiliated To Zhengzhou University, Zhengzhou University, Zhengzhou, Henan Province, China
[2]Department of Anesthesiology, Affiliated Hospital of Guangdong Medical University, Guangdong, China

## AUTHOR ORCIDs

Yan Wang  http://orcid.org/0000-0002-0003-6834
Qinjun Chu  http://orcid.org/0000-0001-9210-5691

## AUTHOR CONTRIBUTIONS

Yan Wang, Formal analysis, Methodology, Project administration, Writing – original draft | Hailong Bing, Data curation, Investigation | Conghui Jiang, Data curation | Jie Wang, Data curation | Xuan Wang, Formal analysis, Methodology, Software | Zhengyuan Xia, Writing – review and editing | Qinjun Chu, Conceptualization, Project administration, Writing – review and editing

## DATA AVAILABILITY

The original contributions presented in the study are publicly available. These data have been assigned at ArrayExpress accession E-MTAB-14351.

## ETHICS APPROVAL

This study was approved by the Regional Ethics Committee of Zhengzhou Central Hospital Affiliated to Zhengzhou University (approval no. 202315). The study participants were included only after obtaining written informed consent. This study was registered in the Chinese Clinical Trial Registry (number ChiCTR2300071050). The patients or volunteers provided their written informed consent to participate in this study.

## ADDITIONAL FILES

The following material is available online.

### Supplemental Material

**Fig. S1 (Spectrum01178-24-s0001.tif).** Differences of gut microbiota.

### Open Peer Review

**PEER REVIEW HISTORY (review-history.pdf).** An accounting of the reviewer comments and feedback.

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
