## [Reviewer comments · Microbiology Spectrum]

Microbiology Spectrum

Gut microbiota dysbiosis and neurological function recovery after intracerebral hemorrhage: An analysis of clinical samples

Yan Wang, Hailong Bing, Conghui Jiang, Jie Wang, Xuan Wang, Zhengyuan Xia, and Qinjun Chu

Corresponding Author(s): Qinjun Chu, Zhengzhou Central Hospital Affiliated to Zhengzhou University

Review Timeline:

Submission Date:	May 12, 2024
Editorial Decision:	July 10, 2024
Revision Received:	August 26, 2024
Accepted:	August 27, 2024

Editor: Jan Claesen

Reviewer(s): Disclosure of reviewer identity is with reference to reviewer comments included in decision letter(s). The following individuals involved in review of your submission have agreed to reveal their identity: Izumi Mashima (Reviewer #1)

Transaction Report:

DOI: <https://doi.org/10.1128/spectrum.01178-24>

Re: Spectrum01178-24 (Gut microbiota dysbiosis and neurological function recovery after intracerebral hemorrhage: An analysis of clinical samples)

Dear Dr. Qinjun Chu:

Thank you for the privilege of reviewing your work. Below you will find my comments, instructions from the Spectrum editorial office, and the reviewer comments.

Thanks for submitting your research to Spectrum. Your work has now been evaluated by two independent Reviewers who are enthusiastic, but have raised some comments to help improve the paper. I would be happy to consider a revised paper which addresses the Reviewer comments in a point-by-point manner (note that Reviewer 1 submitted theirs as an attached pdf document). In addition, both Reviewers had questions about the selection of the IHC cohort/cohorts and how this data was analyzed separately. This would need to be further clarified. Finally, please revise the ASM/Spectrum policy on the data availability and provide the sequencing data in a public repository.

Revision Guidelines

Sincerely,
Jan Claesen
Editor
Microbiology Spectrum

Reviewer #1 (Public repository details (Required)):

Authors must deposit the sequence data to the Sequence Read Archive and show the accession number.

Reviewer #1 (Comments for the Author):

I attached my comments and suggestions as PDF.

Authors should be modify your manuscript mainly statistical analysis, and some parts of discussions didn't support the results. Correct your manuscript carefully.

Reviewer #2 (Public repository details (Required)):

The sequencing data should be deposited and an access number provided.

Reviewer #2 (Comments for the Author):

Wang and colleagues report the gut microbiota composition in patients that recovered from intracerebral hemorrhage and link their data to clinical parameters for neurological recovery. The authors identify a couple of bacterial genera that are potentially associated with IHC and could be predictive for outcome. In general, the paper is ok to follow, but especially the results section could be improved by providing extra context to the experiments and analysis to guide the reader. A major point that the authors need to address is why the patients, who were seemingly all recruited in a similar fashion were parsed into two different cohorts as reported in the manuscript. Why is the analysis not performed on the full IHC versus control cohort and what were the inclusion/exclusion criteria that were used to select the second cohort?

Overall I think this is an interesting study which would be of interest to clinicians treating IHC and I appreciate the authors including a section addressing the limitations of their study at the end of the discussion.

Please find below my comments and suggestions for the authors to consider:

Abstract:

- Line 34: the abstract mentions T1 and T3, but this has not been defined previously. Would recommend defining first or rewording.

Introduction:

- Line 65: not sure alanine is a harmful substance!?

Materials and methods:

- Major: Line 110: Data was collected on the type of postoperative antibiotics provided to the IHC patients, yet this is not addressed elsewhere in the manuscript. It is expected that postoperative antibiotics will have a profound effect on the microbiota composition, and might well account for most of the differences observed at the T3 timepoint. Could the authors provide an analysis into the relative effect of postoperative antibiotics in relation to microbiome changes at T3?

- Line 127: Either fecal samples or rectal swabs were used for the microbiota analysis. What proportion of each sample type was used and did this lead to bias in the data (for example, could you provide beta-dispersion plot distinguishing samples by type)?

Results:

- Line 168: It is surprising that only 30 ICH patients are included in the cohort, why does this not include all 81 patients and what criteria were used to include/exclude patients from this smaller 30-member cohort?

- Line 177 (and throughout): "after ICH". Since there is no longitudinal data available with samples from the patients before they have ICH, it is recommended to reword this to "in ICH patients" since these differences (from the control group) might already have been present in this cohort before the ICH event.

- Line 212: Poor and good groups comes somewhat out of the blue, this could be better introduced/distinguished in the introduction section.

- Line 224: The Anosim (?) analysis was conducted, and the outcome was?

Discussion:

- Line 237: 'genetic diversity' is not assessed with 16S sequencing, please reword.

- Line 242: 'obviously'

- Line 247: 'This study' perhaps 'Future studies will be required to...'

- Line 251-275 reads as a second introduction and does not significantly contribute to the 'discussion' of the results presented.

- Line 298: This increase in Bacteroides in IHC patients is in contrast to the results of the current study.

Conclusion:

- Line 385: 'Aggravated flora disorder'

Availability of data and materials:

- Line 397: The sequencing data should be made available in an appropriate repository and access number provided.

This study investigated that the relationship between the variation of gut microbiota and the neurological function recovery after intracerebral hemorrhage. The focus of this study seems something new and interesting, but some parts of discussion were not supported by the results, and the purpose behind some of the analysis performed in this study were unclear. In addition, the manuscript should be written more politely. Hope my comments and suggestions shown in below help your next manuscript.

Major comments

1. First of all, why the 81 patients were divided to the two cohorts for analysis? Are they all different subjects? Is there any reason to divide two cohorts? All 81 patient's samples should be applied for all analysis in this study. I think it is more evident for this study to use many subjects for one cohort.
2. L104-: About "Data collection", the description of each score is complicated. I recommend showing them as the table and requiring the more polite explanation of each scale or cite the references, because they are important information to determine the prognosis of neurological function in this study.
3. L147-150: Which database did you used to identify the bacteria using OTU? Add the description.
4. L174-178: Authors used the word of "significantly" for Figure 1 results. However, I could not find the statical results of them. If you wanted to use the word of "significantly", the statical results were required.
5. L182-183 & Figure 2C: I could not seem clear difference among the T1, T2, T3 and control groups, because each group were localized both positive and negative direction of PcoA 1 and PcoA 2. In addition, authors conducted the Anosim for the similarity analysis, using Anosim is statistically questionable. Dr. Okansen who is the investigator of vegan recommended using PERMANOVA instead of Anosim, because when the variance between groups differ significantly, problems such as inflated Type 1 and Type 2 errors can occur. I recommend reanalyzing with PERMANOVA and correct the manuscript along with the accurate results.
6. L296-298: This description regarding *Bacteroides* was opposite to the result of this study. Any discussions are required.
7. L317: What is "*Verruca*"? I searched this bacterium using LPSN, but any results were not

shown. Reconfirm it.

8. L343-346: I think the results were not shown for this part of discussion. Show the detail results.
9. Figure 4 and legends: In Figure 4, the results of correlation analysis are shown as heatmap. What are the (A), (B), (C) in the legends? I couldn't see them in Figure 4.
10. Data availability: Authors must deposit the sequence data to the Sequence Read Archive and show the accession number.

Minor comments

1. Bacterial name should be italic. Correct all including figures.
2. L175, 239: *Enterococcus* is the name of genus, correct to *Enterococcaceae*.
3. L200: Add the words, "after surgery" behind "the duration of ICH".
4. L268: "*in vivo*" should be italic.
5. L309: Correct to "phylum *Firmicutes* (update to *Bacillota*)".
6. The phylum names are officially updated. Refer the URL shown in below. Correct all phylum names including figures.
<https://ncbiinsights.ncbi.nlm.nih.gov/2021/12/10/ncbi-taxonomy-prokaryote-phyla-added/>
7. L212-214: Are there the two subtitles? Correct them more understandable.
8. English proof reading is strongly required.

Reviewer #1 (Public repository details (Required)):

Authors must deposit the sequence data to the Sequence Read Archive and show the accession number.

Reviewer #1 (Comments for the Author):

I attached my comments and suggestions as PDF.

Authors should be modify your manuscript mainly statistical analysis, and some parts of discussions didn't support the results.

Correct your manuscript carefully.

This study investigated that the relationship between the variation of gut microbiota and the neurological function recovery after intracerebral hemorrhage. The focus of this study seems something new and interesting, but some parts of discussion were not supported by the results, and the purpose behind some of the analysis performed in this study were unclear. In addition, the manuscript should be written more politely. Hope my comments and suggestions shown in below help your next manuscript.

Major comments

1. First of all, why the 81 patients were divided to the two cohorts for analysis? Are they all different subjects? Is there any reason to divide two cohorts? All 81 patient's samples should be applied for all analysis in this study. I think it is more evident for this study to use many subjects for one cohort.

Thank the reviewer for your careful review. Your extensive academic insight and profound expertise are very admirable. I have thoroughly considered the questions you raised, reviewed the pertinent literature, and subsequently revised the manuscript accordingly.

I concur with your assessment that the results would be more definitive if studied in one cohort. But for the cohort setting of this study, I would like to make an explanation. First, we hope to find the changes of gut microbiota in patients with cerebral hemorrhage and find the differential gut microbiota by comparing patients with healthy people, then verify whether the differential gut microbiota is an independent risk factor for the poor prognosis of patients with cerebral hemorrhage in the second cohort. So in the first cohort, we collected faeces samples from patients with cerebral hemorrhage at three time points, and used the same number of healthy people as the number of patients with cerebral hemorrhage to try to keep the demographic characteristics balanced. In the second cohort, we only collected faeces samples before surgery, but not after surgery, we hoped to be able to find the association between the altered gut microbiota within 24 hours of ICH and patient outcome. So for the second cohort we did not collect more samples.

2. L104-: About "Data collection", the description of each score is complicated. I recommend showing them as the table and requiring the more polite explanation of each scale or cite the references, because they are important information to determine the prognosis of neurological function in this study.

Thank you for your advice, I have added Table 1 and Table 2 to show the OICH score and GOSE score, hoping to explain the meaning of each score more succinctly and clearly.

3. L147-150: Which database did you used to identify the bacteria using OTU? Add the description.

Bacterial and fungal OTU representative sequences were classified taxonomically by blasting against the RDP database and UNITE fungal ITS database, respectively. I have supplemented the description of the database in the manuscript.

4. L174-178: Authors used the word of “significantly” for Figure 1 results. However, I could not find the statistical results of them. If you wanted to use the word of “significantly”, the statistical results were required.

We added Figure S1 to show the statistical differences in gut microbiota between ICH group at T1, T2, T3 and the CON group.

5. L182-183 & Figure 2C: I could not seem clear difference among the T1, T2, T3 and control groups, because each group were localized both positive and negative direction of PcoA 1 and PcoA 2. In addition, authors conducted the Anosim for the similarity analysis, using Anosim is statistically questionable. Dr. Okansen who is the investigator of vegan recommended using PERMANOVA instead of Anosim, because when the variance between groups differ significantly, problems such as inflated Type 1 and Type 2 errors can occur. I recommend reanalyzing with PERMANOVA and correct the manuscript along with the accurate results.

Thank you for your advice, we reperformed the analysis using PERMANOVA and revised the manuscript and Figure 3C for relevant content.

6. L296-298: This description regarding *Bacteroides* was opposite to the result of this study. Any discussions are required.

We have supplemented the manuscript with some discussion of *Bacteroides*. The abundance of *Bacteroides* in the gut microbiota of patients with neurological diseases is controversial. It is difficult to say whether *Bacteroides* have a negative or positive effect on the host. *Bacteroides* are able to benefit the host by preventing infection with potential pathogens that may colonize and infect the gut. However, *Bacteroides*-derived metabolites, carboxylic acids and monosaccharides, may also cause damage to the host. Further studies are needed to explore the role of *Bacteroides* in the gut microbiota and brain-gut axis in ICH patients.

7. L317: What is “Verruca”? I searched this bacterium using LPSN, but any results were not shown. Reconfirm it.

We revised the discussion section in the manuscript to fit the discussion more closely to the results of this study. Inaccurate bacterial names were also modified.

8. L343-346: I think the results were not shown for this part of discussion. Show the detail results.

We have revised the text to address your concerns and hope that it is now clearer. We have completed the revisions, and in the revised manuscript, these modifications are located in L369-375, L379-387.

9. Figure 4 and legends: In Figure 4, the results of correlation analysis are shown as heatmap. What are the (A), (B), (C) in the legends? I couldn't see them in Figure 4.

We were really sorry for our careless mistakes. Thank you for pointing this out. Figure 4 has only one figure, The legends have been revised in the new manuscript.

10. Data availability: Authors must deposit the sequence data to the Sequence Read Archive and show the accession number.

Our submission "Gut microbiota dysbiosis and neurological function recovery after intracerebral hemorrhage [16S rRNA-Seq]" has been assigned ArrayExpress accession E-MTAB-14351.

<https://www.ebi.ac.uk/biostudies/arrayexpress/studies/E-MTAB-14351>

Minor comments

1. Bacterial name should be italic. Correct all including figures.

This question has been revised in the new manuscript.

2. L175, 239: Enterococcus is the name of genus, correct to Enterococcaceae.

Thank you for identifying the error, it has been revised in L33, 192, 258 of the new manuscript.

3. L200: Add the words, "after surgery" behind "the duration of ICH".

It has been revised in L216 of the new manuscript.

4. L268: "in vivo" should be italic.

It has been revised in L420 of the new manuscript.

5. L309: Correct to "phylum Firmicutes (update to Bacillota)".

We thoroughly studied the content of the web page given in the sixth comment, revising the phylum names in the manuscript including figures.

6. The phylum names are officially updated. Refer the URL shown in below. Correct all phylum names including figures.

<https://ncbiinsights.ncbi.nlm.nih.gov/2021/12/10/ncbi-taxonomy-prokaryote-phyla-added/>

In the revised manuscript, these modifications are located in L191.

7. L212-214: Are there the two subtitles? Correct them more understandable.

We have revised the subtitles in the results to make them clearer , these modifications are located in L228,236 in the new manuscript.

8. English proof reading is strongly required.

We would like to thank the referee again for taking the time to review our manuscript. We have tried our best to polish the language in the revised manuscript, and the revised manuscript has been proofread by a native English speaker.

Reviewer #2 (Public repository details (Required)):

The sequencing data should be deposited and an access number provided.

Our submission "Gut microbiota dysbiosis and neurological function recovery after intracerebral hemorrhage [16S rRNA-Seq]" has been assigned ArrayExpress accession E-MTAB-14351. <https://www.ebi.ac.uk/biostudies/arrayexpress/studies/E-MTAB-14351>

Reviewer #2 (Comments for the Author):

Wang and colleagues report the gut microbiota composition in patients that recovered from intracerebral hemorrhage and link their data to clinical parameters for neurological recovery. The authors identify a couple of bacterial genera that are potentially associated with IHC and could be predictive for outcome. In general, the paper is ok to follow, but especially the results section could be improved by providing extra context to the experiments and analysis to guide the reader. A major point that the authors need to address is why the patients, who were seemingly all recruited in a similar fashion were parsed into two different cohorts as reported in the manuscript. Why is the analysis not performed on the full IHC versus control cohort and what were the

inclusion/exclusion criteria that were used to select the second cohort?

Overall I think this is an interesting study which would be of interest to clinicians treating IHC and I appreciate the authors including a section addressing the limitations of their study at the end of the discussion.

Thank the reviewer for your careful review. Your extensive academic insight and profound expertise are very admirable. I have thoroughly considered the questions you raised, reviewed the pertinent literature, and subsequently revised the manuscript accordingly.

For the cohort setting of this study, I would like to make an explanation. First, we hope to find the changes of gut microbiota in patients with cerebral hemorrhage and find the differential gut microbiota by comparing patients with healthy people, then verify whether the differential gut microbiota is an independent risk factor for the poor prognosis of patients with cerebral hemorrhage in the second cohort. So in the first cohort, we collected faeces samples from patients with cerebral hemorrhage at three time points, and used the same number of healthy people as the number of patients with cerebral hemorrhage to try to keep the demographic characteristics balanced. In the second cohort, we only collected faeces samples before surgery, but not after surgery, we hoped to be able to find the association between the altered gut microbiota within 24 hours of ICH and patient outcome. So for the second cohort we did not collect more samples, and the inclusion/exclusion criteria were the same for all ICH patients in this study.

Please find below my comments and suggestions for the authors to consider:

Abstract:

- Line 34: the abstract mentions T1 and T3, but this has not been defined previously.

Would

recommend defining first or rewording.

Thank you for bringing this to our attention, we have revised the abstract part and explained the T1, T2, T3 time points in line 22-23.

Introduction:

- Line 65: not sure alanine is a harmful substance!?

This quote from the original reference, which may be controversial, has been deleted

Materials and methods:

- Major: Line 110: Data was collected on the type of postoperative antibiotics provided to the IHC

patients, yet this is not addressed elsewhere in the manuscript. It is expected that postoperative

antibiotics will have a profound effect on the microbiota composition, and might well account for

most of the differences observed at the T3 timepoint. Could the authors provide an analysis into

the relative effect of postoperative antibiotics in relation to microbiome changes at T3?

we acknowledge the significance of this aspect, antibiotic administration has been shown to alter the structure and function of the gut microbiome. Animal studies have demonstrated that long-term modification of gut microbiota by broad-spectrum antibiotics can improve stroke outcomes in rats. Therefore, this study collected the postoperative antibiotic use of patients with ICH. However, during the clinical data collection process, it was observed that patients in the ICU or surgical ward were often treated with a variety of antibiotics, frequently in combination. This complexity posed significant challenges to data collection and analysis, rendering definitive conclusions difficult to ascertain. It was challenging to determine the specific impact of antibiotics on the gut microbiota. We can only observe the changes of gut microbiota after surgery based on the real world, but we did not evaluate the relative effects of postoperative antibiotics on microbiome alterations at T3. Additionally, in this study, preoperative changes in intestinal flora were utilized to predict neurological function outcomes 30 days post-surgery, with the aim of minimizing antibiotic interference. And in the limitations of the article, we also explain the antibiotics, the effects of different types of antibiotics and their use on the gut microbiota in vivo are not clear.

- Line 127: Either fecal samples or rectal swabs were used for the microbiota analysis. What proportion of each sample type was used and did this lead to bias in the data (for example, could you provide beta-dispersion plot distinguishing samples by type)? Thank you for your advice, but we did not count the proportion of fecal samples and anal swabs, but studies have found that rectal swab samples and fecal samples of healthy volunteers and patients were taken, DNA was extracted, 16S rDNA sequencing was performed, and the data were analyzed and compared. Although there were small differences between the two samples, which reflected differences across body parts, the overall results were consistent and reproducible(1). So we think there should be no bias.

1. Biehl LM, Garzetti D, Farowski F, Ring D, Koeppel MB, Rohde H, Schafhausen P, Stecher B, Vehreschild M. 2019. Usability of rectal swabs for microbiome sampling in a cohort study of hematological and oncological patients. *PLoS one* 14: e0215428.

Results:

- Line 168: It is surprising that only 30 ICH patients are included in the cohort, why does this not include all 81 patients and what criteria were used to include/exclude patients from this smaller 30-member cohort?

Regarding the setup of the study cohort, we have explained it above. With regard to RNA-Seq, important experimental design decisions include the choice of sequencing depth and number of biological replicates to use. The sample number of 16S sequencing suggests that human intestinal and fecal samples vary greatly from individual to individual, and it is recommended that each group be more than 30 biological replicates. Therefore, faeces samples were collected at three time points from 30 ICH patients in the first cohort and were used as controls with 30 healthy volunteers.

- Line 177 (and throughout): "after ICH". Since there is no longitudinal data available with samples from the patients before they have ICH, it is recommended to reword this to "in ICH patients" since these differences (from the control group) might already have been present in this cohort before the ICH event.

Thank you for your advice, we have revised the relevant content in the new manuscript.

- Line 212: Poor and good groups comes somewhat out of the blue, this could be better introduced/distinguished in the introduction section.

Thank you for your advice, we have revised the relevant content in the new manuscript, the patients were divided into POOR group or GOOD group according to the GOS-E score of patients with ICH 30 days after surgery. And in the materials and methods section, the GOS-E score is explained in detail through Table 2.

- Line 224: The Anosim (?) analysis was conducted, and the outcome was?
At the suggestion of another reviewer, we re-performed the analysis using PERMANOVA test instead of Anosim analysis and present the results in the L241 of the new manuscript.

Discussion:

- Line 237: 'genetic diversity' is not assessed with 16S sequencing, please reword.
Thank you for your advice, we have made major revisions to the discussion section, corrected some inappropriate expressions.

- Line 242: 'obviously'
We have revised this inappropriate expression.

- Line 247: 'This study' perhaps 'Future studies will be required to...'
Thank you for your advice, we revised the sentence to: “Future studies will be required to explore the mechanism of Lachnospiraceae in enhancing the prognosis of patients with ICH”.

- Line 251-275 reads as a second introduction and does not significantly contribute to the
'discussion' of the results presented.

We modified this paragraph to reduce the discussion unrelated to the results in the L275-286 of new manuscript.

- Line 298: This increase in *Bacteroides* in IHC patients is in contrast to the results of the current
study.

We have supplemented the manuscript with some discussion of *Bacteroides* in the L320-337 of new manuscript. The abundance of *Bacteroides* in the gut microbiota of patients with neurological diseases is controversial. It is difficult to say whether *Bacteroides* have a negative or positive effect on the host. *Bacteroides* are able to benefit the host by preventing infection with potential pathogens that may colonize and infect the gut. However, *Bacteroides*-derived metabolites, carboxylic acids and monosaccharides, may also cause damage to the host. Further studies are needed to explore the role of *Bacteroides* in the gut microbiota and brain-gut axis in ICH patients.

Conclusion:

- Line 385: 'Aggravated flora disorder'
We have revised this inappropriate expression in the L430 of new manuscript.

Availability of data and materials:

- Line 397: The sequencing data should be made available in an appropriate repository and access number provided.

We would like to thank the referee again for taking the time to review our manuscript. Our submission "Gut microbiota dysbiosis and neurological function recovery after intracerebral hemorrhage [16S rRNA-Seq]" has been assigned ArrayExpress accession E-MTAB-14351. <https://www.ebi.ac.uk/biostudies/arrayexpress/studies/E-MTAB-14351>.

Re: Spectrum01178-24R1 (Gut microbiota dysbiosis and neurological function recovery after intracerebral hemorrhage: An analysis of clinical samples)

Dear Dr. Qinjun Chu:

Thank you for carefully addressing the Reviewers' comments. I would hereby like to congratulate you on the acceptance of your manuscript for publication in Spectrum!

Your manuscript has been accepted, and I am forwarding it to the ASM production staff for publication. Your paper will first be checked to make sure all elements meet the technical requirements. ASM staff will contact you if anything needs to be revised before copyediting and production can begin. Otherwise, you will be notified when your proofs are ready to be viewed.

Sincerely,
Jan Claesen
Editor
Microbiology Spectrum